# Clinical Features of Gastric Signet Ring Cell Cancer: Results from a Systematic Review and Meta-Analysis

**DOI:** 10.3390/cancers15215191

**Published:** 2023-10-28

**Authors:** Mariagiulia Dal Cero, Maria Bencivenga, Drolaiz H. W. Liu, Michele Sacco, Mariella Alloggio, Kelly G. P. Kerckhoffs, Federica Filippini, Luca Saragoni, Mar Iglesias, Anna Tomezzoli, Fátima Carneiro, Heike I. Grabsch, Giuseppe Verlato, Lorena Torroni, Guillaume Piessen, Manuel Pera, Giovanni de Manzoni

**Affiliations:** 1General and Upper GI Surgery Division, Department of Surgery, University of Verona, Borgo Trento Hospital, Piazzale Stefani 1, 37124 Verona, Italy; 1325220@uab.cat (M.D.C.);; 2Section of Gastrointestinal Surgery, Hospital Universitario del Mar, Hospital del Mar Medical Research Institute (IMIM), Department of Surgery, Universitat Autònoma de Barcelona, 08003 Barcelona, Spain; 3Department of Pathology, GROW School for Oncology and Reproduction, Maastricht University Medical Center, 6229 HX Maastricht, The Netherlands; 4Institute of Clinical Pathology and Molecular Pathology, Kepler University Hospital and Johannes Kepler University, 4021 Linz, Austria; 5Department of Pathology, VieCuri Medical Centre, 5912 BL Venlo, The Netherlands; 6Pathology Unit, Morgagni-Pierantoni Hospital, 47100 Forlì, Italy; 7Department of Pathology, Hospital Universitario del Mar, Hospital del Mar Medical Research Institute (IMIM), 08003 Barcelona, Spain; 8Department of Pathology, Verona University Hospital, 37134 Verona, Italy; 9Department of Pathology, Medical Faculty of the University of Porto/Centro Hospitalar Universitário São João and Ipatimup/i3S, 4200-319 Porto, Portugal; 10Division of Pathology and Data Analytics, Leeds Institute of Medical Research at St. James’s, University of Leeds, Leeds LS2 9JT, UK; 11Unit of Epidemiology and Medical Statistics, Department of Diagnostics and Public Health, University of Verona, 37126 Verona, Italy; 12Department of Digestive and Oncological Surgery, Lille University Hospital, 59000 Lille, France

**Keywords:** gastric cancer, signet ring cells, survival, poorly cohesive, pathological classification

## Abstract

**Simple Summary:**

The clinical behaviour of signet ring cell histology in gastric cancer has long been a subject of controversy. Recent years have underscored the pressing issue of a lack of a standardised definition for signet ring cell histology, leading to its often-ambiguous placement within broader categories associated with poor prognosis. Conversely, comparisons of signet ring cell gastric cancer have been made against a wide spectrum of non-signet ring cell cases, introducing significant heterogeneity. The primary objective of this literature search and subsequent meta-analysis was to gain a deeper understanding of signet ring cell gastric cancer. Our findings revealed that the prognosis of signet ring cell gastric cancer is intricately tied to the disease stage, yet it is also contingent on the specific comparison group employed. The variability in signet ring cell cancer’s clinical behaviour may stem from the absence of a standardised definition. Therefore, it is imperative to work towards a uniform classification system for gastric cancer to enhance clarity and coherence in future research and clinical practice.

**Abstract:**

Background: Conflicting results about the prognostic relevance of signet ring cell histology in gastric cancer have been reported. We aimed to perform a meta-analysis focusing on the clinicopathological features and prognosis of this subgroup of cancer compared with other histologies. Methods: A systematic literature search in the PubMed database was conducted, including all publications up to 1 October 2021. A meta-analysis comparing the results of the studies was performed. Results: A total of 2062 studies referring to gastric cancer with signet ring cell histology were identified, of which 262 studies reported on its relationship with clinical information. Of these, 74 were suitable to be included in the meta-analysis. A slightly lower risk of developing nodal metastases in signet ring cell tumours compared to other histotypes was found (especially to undifferentiated/poorly differentiated/mucinous and mixed histotypes); the lower risk was more evident in early and slightly increased in advanced gastric cancer. Survival tended to be better in early stage signet ring cell cancer compared to other histotypes; no differences were shown in advanced stages, and survival was poorer in metastatic patients. In the subgroup analysis, survival in signet ring cell cancer was slightly worse compared to non-signet ring cell cancer and differentiated/well-to-moderately differentiated adenocarcinoma. Conclusions: Most of the conflicting results in signet ring cell gastric cancer literature could be derived from the lack of standardisation in their classification and the comparison with the different subtypes of gastric cancer. There is a critical need to strive for a standardised classification system for gastric cancer, fostering clarity and coherence in the forthcoming research and clinical applications.

## 1. Introduction

Gastric cancer (GC) is a complex and heterogeneous disease. Despite its declining incidence in developed countries, such as the USA, it remains the fourth leading cause of cancer-related mortality [1]. Signet ring cell carcinoma (SRCC) is a subtype of poorly cohesive gastric cancer, which can be challenging to diagnose with conventional methods [2], and accurate disease staging can be complicated. An explorative laparoscopy is required to detect peritoneal carcinomatosis, which is more common in poorly cohesive cancers and usually not detectable on a CT scan [3,4,5]. It has been suggested that the prognosis of patients with signet ring cell (SRC) gastric cancer depends on the disease stage as for other histological subtypes [6]. Whether the percentage of SRC within the tumour may predict survival and response to preoperative therapy is still a matter of debate [7,8].

SRCC was described as a histological subtype of gastric cancer in the 1st edition (1977) of the World Health Organization (WHO) classification [9] and was defined as a tumour predominantly or exclusively composed of signet ring cells [10]. Since the publication of the 4th edition of the WHO classification (2010), SRCC has been considered a subtype of poorly cohesive carcinoma [11]. In the recent 5th edition of the WHO classification, poorly cohesive (PC) gastric cancer was subdivided into the SRCC subtype (>90% SRC) and PC not otherwise specified (PC-NOS) [12]. The Japanese Gastric Cancer Association (JGCA) classification originally classified SRCC as undifferentiated adenocarcinoma. Since the 2nd edition (1993), poorly differentiated adenocarcinoma (PDA) has been subdivided into solid and non-solid subtypes [13]. The non-solid subtype PDA corresponds to the poorly cohesive subtype of the WHO classification and the diffuse type in the Laurèn classification [11]. In the Laurèn classification, SRCC and other poorly cohesive GCs are classified as “diffuse”-type GCs [14]. The Nakamura classification includes SRCC in the “undifferentiated” category [15]; see Figure 1.

A significant difference between the classifications most commonly used in the West (WHO and Laurèn) and those used in the East (JGCA, Nakamura) is that, in the West, there is a concept of ‘poorly cohesive’ cancer. In the East, there is a concept of ‘poorly differentiated’ cancer [12]. Unfortunately, these two concepts do not fully overlap, causing difficulties in interpreting study results with subsequent knowledge gaps.

Generally, GC should be composed predominantly or exclusively of SRC to be classified as SRCC. According to several authors, SRCC is an adenocarcinoma in which more than 50% of the tumour consists of isolated or small groups of malignant cells containing intra-cytoplasmic mucins [11]. Despite this, a universal standardised definition of SRCC is yet to be found, and, frequently, it is not clear which criteria are used to classify SRCC.

Many studies including clinical trials lump SRCC with other subtypes; therefore, this study aims to investigate the use of different definitions and histopathological classifications of SRCC through a comprehensive literature review and meta-analysis. We will further analyse the relationship between pathological classifications with prognosis and treatment outcome.

## 2. Materials and Methods

### 2.1. Search Strategy and Study Selection

This literature review and meta-analysis were conducted following the PRISMA guidelines [16]. The study was not registered in PROSPERO.

A comprehensive literature search was performed using the PubMed database, including articles published in English from 1947 to 1 October 2021, using synonyms and Medical Subject Headings (MeSH) terms for ‘gastric’ and ‘signet-ring cell cancer’ (Appendix A). Two authors (MDC and LT) independently conducted the search and identification of manuscripts that could be included in this study by screening publication titles and abstracts, while a third author (MB) checked any disagreement and confirmed that the selected manuscripts met the inclusion criteria.

We included studies that reported on clinical aspects of SRCC in gastric cancer patients. We excluded studies reporting (1) results from less than 10 SRCC cases; (2) case reports referring to hereditary diffuse gastric cancer (HDGC); (3) cell culture-based studies or animal studies reporting only on oesophageal adenocarcinomas; and (4) studies where we were unable to retrieve the full-text version of an article. We also excluded studies focussing on histopathological aspects as our group published these studies separately [17].

We finally refined the selection, detecting papers that contained useful data for the meta-analysis, such as the lymph node metastasis and lymphovascular invasion and survival hazard ratio (HR) or relative risk (RR) in SRCC compared to other common gastric cancer histotypes. All the definitions of SRC were accepted; nevertheless, to reduce the variability, we eventually excluded former studies (before 2010, when the category of poorly cohesive gastric cancer was introduced [11]) and studies that utilised non-comparable groupings (for instance, papers comparing pure SRCC to SRCC with 50–100% or 10–90% SRC; papers where SRCCs were grouped together with PDA; papers where SRCCs were compared to uncommon gastric cancer as adenosquamous or hepatoid; papers focussing on node-negative or synchronous multifocal gastric cancer).

The following study characteristics were recorded: name of the first author, year of publication, country and continent of patient cohort origin, stage of gastric disease cancer, the total number of patients included in the study, number of SRCC and non-signet ring cell carcinoma (NSRCC) patients, histopathological classification used, % of patients with lymph node metastasis, presence of lymphovascular invasion, and relationship between histological phenotype and survival. The observational studies were evaluated to assess the risk of bias using the Newcastle–Ottawa scale (NOS) [18], a scale designed to assess the quality of nonrandomised studies in interpreting meta-analytic results.

### 2.2. Analysis of Histopathologic Classification Systems and Definitions of SRCC

For each of the studies included, we first analysed which histopathological classification was used and whether a specific definition of SRC was provided. We reported the histopathologic classification systems and definitions of SRCC used in the different studies according to the reference systems reported above [12,13,14].

### 2.3. Statistical Analysis—Meta-Analysis

A meta-analysis was conducted in the selected studies, comparing the results of the various studies and focusing on lymph node and lymphovascular invasion and survival, which are important features in the prognosis of gastric cancer. The RR was calculated for categorical variables (lymph node and lymphovascular invasion), and the HR for time-dependent variables (survival). To evaluate the variability among studies, we computed a heterogeneity test and the I^2^ statistic, indicating the proportion of total variation among the effect estimates of different studies attributed to heterogeneity rather than sampling error. When the heterogeneity test was not significant (*p* > 0.050), I^2^ was less than 30% [19,20], and significant heterogeneity was ruled out. In this case, a fixed-effects model was adopted to evaluate the results pooled using the method of Mantel and Haenszel. Otherwise, a random effects model was used, and the pooling of results was performed using the DerSimonian and Laird method [21]. Egger’s test and the funnel plot addressed the small study effect.

The level of statistical significance was set at 5%, and confidence intervals (CI) were calculated at 95%. The results were displayed graphically using forest plots.

The STATA software, version 17 (StataCorp, College Station, TX, USA), was used for the analysis.

## 3. Results

The literature search in PubMed (Figure 2) resulted in 2062 articles published between 1947 and 1 October 2021. We excluded 1799 articles: 693 case reports, 493 focused on pathological aspects, 403 were not relevant, 85 cell culture-based studies or animal studies, 75 on HDGC, and 33 with less than ten or no SRCC cases, 14 did not have full-text availability, and 4 focused on oesophageal carcinoma. In total, 262 articles were included.

By searching for papers that contained useful data for the meta-analysis and excluding studies from before 2010, we finally included 74 papers. We first evaluated these studies according to the Newcastle–Ottawa scale: 19 scored six points (26%), 25 scored seven points (34%), 23 scored eight points (31%), and 7 scored nine points (9%) (Appendix A). Since we consider the quality threshold of six points, all 74 studies were included in the meta-analysis (Figure 2). The general characteristics of the included studies are presented in Table 1 and, more specifically, in Appendix A.

In the set of 74 selected articles, we searched for the definition of SRCC and the pathological classification used, and we found that both varied between studies (Figure 3). The WHO classification [12] was most frequently used (27 studies—37%), 7 studies (10%) used the Japanese classification [13], and 1 study (1%) used Laurèn’s classification [14]. A total of 24 studies (32%) used more than one classification (WHO, Japanese, Laurèn, Nakamura), and 15 studies (20%) did not specify the classification used. Only 1 (1%) study [38] used the term “poorly cohesive carcinoma” as proposed in the 4th and 5th edition of the WHO classification [11,12].

In the whole cohort of studies, 1 study (1%) cited the 1st edition of the WHO classification, 16 (22%) the 2nd, 3 (4%) the 3rd, 16 (22%) the 4th, 2 (3%) the 5th. In parallel, 1 study (1%) cited the 1st edition of the Japanese classification, 6 (8%) the 2nd, 10 (11%) the 3rd, and 1 (1%) the 4th. Furthermore, 23 studies (31%) did not specify the classification edition used. Indeed, seven studies (9%) used information from the SEER database and, therefore, with a probable lack of specific clinical data, citied different editions of the WHO classification. Five studies (7%) only mentioned the type of classification in the introduction of the manuscript without specifying its use in the study.

### 3.1. Lymph Node Metastasis

Among all of the studies reporting data about lymph node metastasis (LNM) in all stages of gastric cancer, 15 papers compared SRCC to NSRCC, 11 compared SRCC to differentiated carcinoma/well-to-moderately differentiated adenocarcinoma/adenocarcinoma not otherwise specified (DC/WMDA/ADK), 16 to undifferentiated carcinoma/poorly differentiated adenocarcinoma (UDC/PDA)/mucinous, and 2 to mixed adenocarcinoma. Among the studies reporting data about lymph node metastasis in early gastric cancer (EGC), 7 papers compared SRCC to NSRCC, 14 to DC/WMDA/ADK, 21 to UDC/PDA/mucinous, and 5 to a mix of cancers. Among the studies reporting data about lymph node metastasis in advanced gastric cancer (AGC), 5 papers compared SRCC to NSRCC, 6 to DC/WMDA/ADK, and 6 to UDC/PDA/mucinous.

Figure 4A shows the RR of lymph node metastasis considering all stages of gastric cancer. The forest plot comparing SRCC to other histologies showed a slightly lower risk of developing nodal metastases (RR = 0.93, 95% CI 0.87–0.98) with high variability across studies (I^2^ = 94.8%). When the histotype of controls stratified the analysis, a clear pattern emerged. Indeed, the risk of nodal metastases in SRCC was similar compared to NSRCC, lower compared to the UDC/PDA/mucinous and mixed group, or higher compared to the DC/WMDA/ADK group. Also, the variability across studies slightly decreased.

Stratification of the studies according to the tumour stage showed that, in early gastric cancer, the RR of lymph node metastasis was clearly lower in SRCC compared to other histologies (RR = 0.68, 95% CI 0.58–0.79, see Figure 4B). The variability across studies, although still significant, markedly decreased (I^2^ was 77.5%). The pattern of the risks of nodal metastases in SRCC compared to other histotypes was overlapping in the early and all stages of SRCC with a shift to the left (with an overall lower risk of lymph node metastasis in SRCC).

Conversely, in advanced gastric cancer, the risk of nodal metastases in SRCC slightly increased (RR = 1.05, 95% CI 1.00–1.10, see Figure 4C). Again, variability across studies decreased (I^2^ was 61.9%).

### 3.2. Lymphovascular Invasion

A similar pattern was found for lymphovascular invasion (LVI) (see Appendix A).

### 3.3. Survival

Figure 5A shows the comparison of multivariable HR/RR of survival of SRCC with other histotypes divided with regard to NSRCC (11 studies), DC/WMDA/ADK (4 studies), and UDC/PDA/mucinous (4 studies). The prognosis of SRC tumours was significantly worse than the other tumours (RR of mortality in SRCC versus controls = 1.16, 95% CI 1.07–1.24). SRCC survival was similar compared to UDC/PDA/mucinous, and slightly worse than NSRCC and DC/WMDA/ADK. The variability across studies was high (I^2^ = 84.6%). However, a qualitative interaction was observed between SRCC and tumour stage. Figure 5B shows survival for the early (9 studies, Guo CG has two HR because of the comparison of SRCC with the DC/WMDA/ADK group and the UDC/PDA/mucinous group), advanced (8 studies), and metastatic stages (6 studies). Compared to other histotypes, SRCC tumours had better survival in the early stages (RR = 0.67, 95% CI 0.38–0.97), a similar survival in the advanced stages (RR = 1.15, 95% CI 0.94–1.36), and worse survival in metastatic cancers (RR = 1.29, 95% CI 1.09–1.49).

Publication Bias: The publication bias was evaluated with the funnel plot and Egger’s test. No publication bias was found on early stage gastric cancer (*p* = 0.294) or on all-stage gastric cancer (*p* = 0.861) in the survival outcomes. The funnel plot and Eggers’ test are shown in the Appendix A.

## 4. Discussion

SRC gastric carcinoma is a subtype of gastric cancer with peculiar and controversial characteristics. At first, SRCC was described as a tumour with a poor prognosis; however, the growing available literature recently affirmed that when SRCC is found at early stages, limited to the gastric mucosa or submucosa, it has a better prognosis than all the other GC subtypes. Still, when it progresses through the gastric wall, it becomes highly aggressive, carrying high rates of nodal metastases and peritoneal carcinomatosis [3,4,5,6]. Nevertheless, the wide heterogeneity of definitions and classifications of SRCC contributes to the great confusion about its behaviour. Moreover, the authors’ grouping of patients affected with SRCC with PDA or with other diffuse/poorly cohesive carcinomas decreases the homogeneity in the literature even more. Therefore, a pivotal step to more robust evidence is standardising the terminology used to define this cancer subtype.

Our study clarifies that the pathological classification used in the papers on gastric cancer and the definition of SRCC adopted in different studies are highly variable. We should pay attention to what is meant by SRCC in the different studies, and it is also important to what SRCC is compared with, that is, what is meant by NSRCC (some studies compare SRCC with well-differentiated tumours, while others with poorly differentiated tumours, which are still not as a type of SRCC). Therefore, the results of the studies found in this extensive literature review about SRCC have a great variability for this reason. Moreover, a particularly poor prognosis could be added if SRCC patients are grouped as diffused (Lauren classification) or undifferentiated (Nakamura and Japanese classifications). In fact, this would mean grouping SRCC with the poorly cohesive type 2 class (considering the Japanese classification) or to the poorly cohesive NOS class (considering the WHO classification). Only a few studies analysed the group poorly cohesive type 2 (Japanese classification)/poorly cohesive NOS (WHO classification) classes separately; therefore, this subtype is likely sometimes analysed with SRCC, and some others with NSRCC, which adds variability (see Figure 1). As such, there is a need to universalise the histopathological definitions used worldwide to allow for a more homogeneous comparison between the subgroups of gastric cancer.

This study’s main result is that the prognosis of SRCC tumours largely depends on the stage of cancer, as confirmed by the other three meta-analyses that started investigating this topic. In fact, two of these three meta-analyses showed superimposable results, agreeing that lymph node metastasis was lower in ECG in SRCC compared to NSRCC, without differences in AGC [94,95]. Regarding survival, SRCC was associated with poorer overall survival when analysing all stages of gastric cancer [96], although this was not always statistically significant [94,95]. In EGC, the subgroup analysis showed better survival in two meta-analyses [95,96] and comparable survival outcomes in the other one [94]. Regarding AGC, one of the studies showed worse survival in SRCC when excluding patients with metastases. However, this sub-analysis evaluated only three studies, and the analysis, including metastatic patients, did not show statistical differences [96]. Another study demonstrated a worse prognosis in the advanced stage but did not separate patients with stage IV cancer [95]. The last meta-analysis also showed worse survival in the advanced tumour stage. No significant difference in survival outcomes was demonstrated in patients with metastases [94].

In our study, regarding lymph node invasion and considering all stages of gastric cancer, we found a slightly lower risk to develop nodal metastases in SRCC. The subgroup analysis comparing SRCC with different histotypes added the evidence that the risk of nodal metastases in SRCC tumours was lower, similar, or higher when compared to UDC/PDA/mucinous, mixed, NSRCC, and to DC/WMDA/ADK respectively. In EGC, as demonstrated by the other two meta-analysis [94,95], SRCC was associated with a lower incidence of LNM. In the subgroup analysis, we noticed that the incidence of LNM was lower, especially when SRCC was compared to UDC/PDA/mucinous and mixed, rather than being similar when SRCC was compared to NSRCC and to DC/WMDA/ADK. The analysis of lymphovascular invasion highlighted a similar trend.

Regarding survival, all stages of SRCC had worse prognostic outcomes than other histotypes. The separate analysis regarding the stage of cancer confirmed the evidence shown by the previous meta-analysis with a better prognosis in early SRCC than in other histotypes, a similar prognosis in advanced cancer (3 of the 8 studies included patients with stage IV cancer in the advanced group), and worse in patients with metastases. When sorting the studies by histologies, the worse prognostic outcome of SRCC was confirmed, especially when comparing SRCC to NSRCC and to DC/WMDA/ADK, while no difference was seen in the comparison to UDC/PDA/mucinous.

Our analysis highlights the importance of identifying the comparison histologic group, which might provide significant variability in relation to the results. In addition, it confirms the different prognoses of SRCC based on stage—evidence that is becoming more and more clear. Indeed, these tumours tend to be more aggressive compared to other subtypes as they become more advanced, and this behaviour could be explained by the impact of peritoneal carcinomatosis. Patients with SRCC, in fact, are at an especially high risk for an occult misdiagnosed peritoneal disease, as it was shown by several studies [3,4,5]. A recent study found that peritoneal lavage during exploratory laparoscopy was positive in about 32.1% of the patients with SRCC cM0 [4]. Also, patients with gastric SRCC that are submitted to curative resection are at a risk of developing peritoneal metastasis [97,98,99,100]; recurrence was reported to occur in 51% of patients after a 54-month follow up, with 19% peritoneal recurrence [101]. Moreover, in SRCC with peritoneal metastasis, the prognosis appears to remain poor, irrespective of whichever treatment is used. Chemotherapy is less effective than other histologies and has a shorter survival rate [77]. Cytoreductive surgery, intraoperative hyperthermic intraperitoneal chemotherapy (CRS+HIPEC), and low-dose pressurised intraperitoneal aerosol chemotherapy (PIPAC) are techniques which are still under investigation but can be used; however, they should be restricted to highly selective patients [49,102,103].

This study, while providing valuable insights, still contains some limitations. Firstly, the quality of the meta-analysis heavily relies on the quality of the included studies. If the individual studies exhibit biases or methodological flaws, these can propagate into the meta-analysis, compromising the overall reliability of the findings. Moreover, the heterogeneity among the selected studies, such as variations in patient populations, diagnostic criteria (use of EUS/PET/diagnostic laparoscopy), and treatment modalities (such as multimodal treatment), can pose a challenge in drawing cohesive conclusions. Additionally, this meta-analysis might lack the ability to capture the recent advancements in diagnostic techniques or treatments due to the inclusion of studies conducted over extended periods of time.

To encourage clinicians to offer a correct diagnosis, a recent expert consensus proposed a new classification system in which only PC carcinomas with more than 90% of SRC should be classified as SRCC. PC non-signet ring cell carcinomas should be further subdivided into PC carcinomas with a SRC component (10–90% SRC) and PC carcinomas not otherwise specified (< 10% SRC) [104]. This classification seems to have practical prognostic relevance [105,106]; in fact, confusing poorly cohesive tumours with different percentages of SRC could cause conflicting findings in the literature concerning the prognosis and the response to chemo-radiotherapy. Of note, this classification of PC carcinomas based on the amount of SRC has also been cited by the very last edition of the World Health Organization (WHO) classification that divides PC gastric cancer into SRCC (>90%) and PC non-otherwise specified (PC-NOS) [12]. The incidence of this subtype of tumour is increasing with a rise of young patients affected and a poor prognosis due to its rapid progression, high rate of peritoneal disease, and high recurrence, even after a curative resection and early diagnosis. A correct definition and clear histological comparison of the disease are of primary concern, as the results of the diagnosis can completely change based on this information. Consequently, this would help in clarifying the prognosis and developing a correctly tailored treatment for each patient.

## 5. Conclusions

In conclusion, most of the conflicting results found by analysing the available literature on SRCC could come from the lack of standardisation of its pathological definition and classification that is being used. Therefore, we suggest the use of a clear classification of SRCC, in addition to considering the percentage of SRC when investigating this type of tumour [104].

From the results of this meta-analysis, we concluded that SRCC has a lower tendency to lymph node invasion in the early stage of cancer, especially in comparison with UDC/PDA/mucinous and mixed tumours. This difference decreases with tumour progression, as advanced SRCC tends to have similar lymph node metastasis as the other histotypes. Survival tends to be greater in early SRCC than in EGC of the other histotypes. At the same time, it is worse when all stages are compared together, likely reflecting the dramatic impact of peritoneal involvement in advanced SRCC compared to the other histologies.

Over the years, there has been a trend towards a uniformity of data, probably thanks to a more precise and homogeneous pathological diagnosis. Hopefully, this harmonisation process will continue with the final aim of achieving a clearer prognosis and ameliorating the treatment of patients with SRCC.

## Figures and Tables

**Figure 1 cancers-15-05191-f001:**
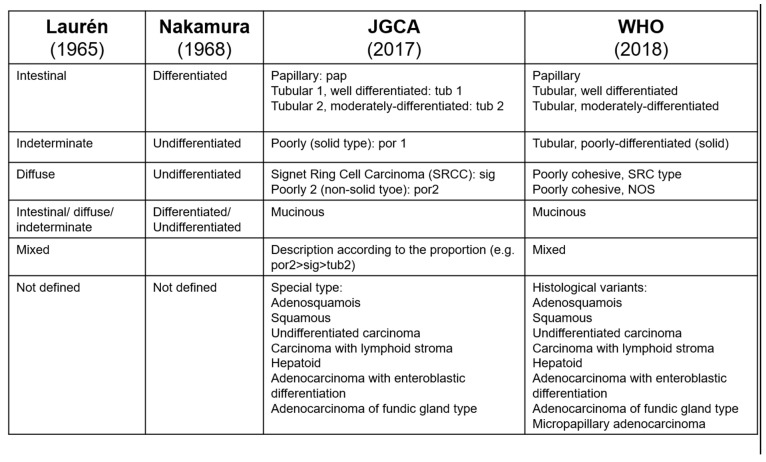
Pathological classifications of gastric cancers.

**Figure 2 cancers-15-05191-f002:**
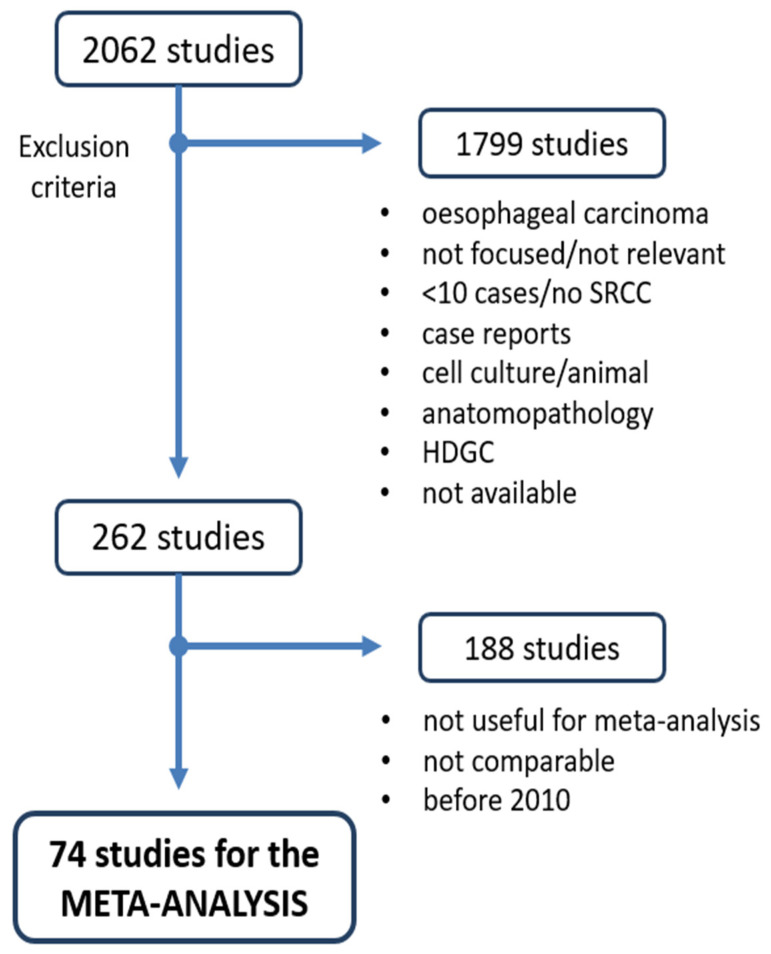
Flow chart of the literature search and selection of studies.

**Figure 3 cancers-15-05191-f003:**
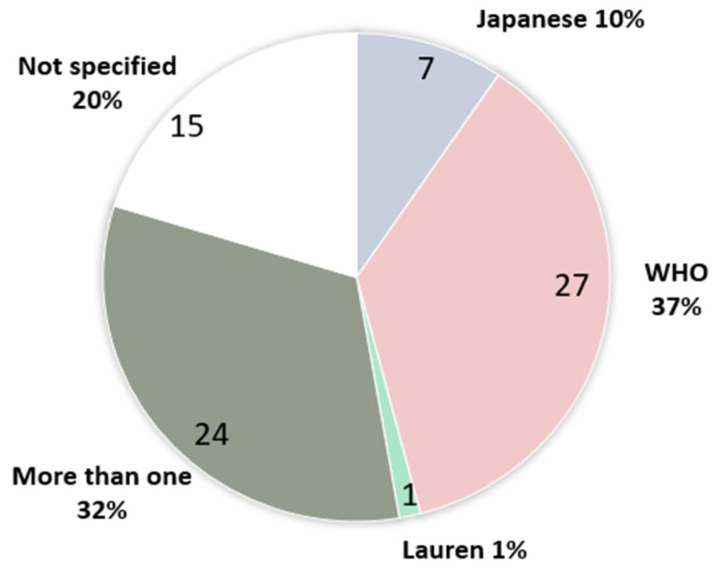
Use of pathological classification in gastric cancer.

**Figure 4 cancers-15-05191-f004:**
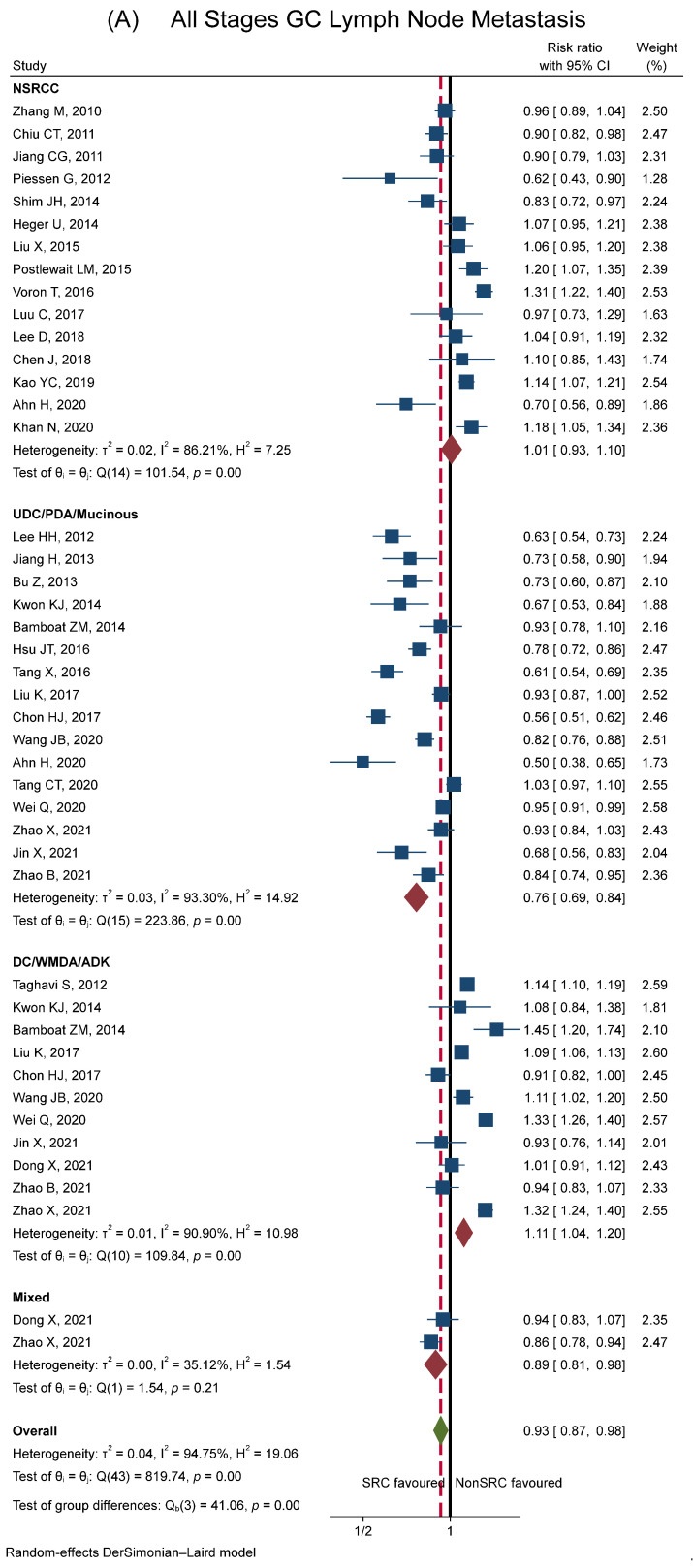
Forest plot showing meta-analysis results for lymph node metastasis between SRCC and other histologies (NSRCC, UDC/PDA/mucinous, DC/WMDA/ADK, mixed). Plot (**A**): patients with all stages of gastric cancer. Plot (**B**): patients with early gastric cancer. Plot (**C**): patients with advanced gastric cancer. The blue square represents the size proportional to the weight of the study, and the horizontal lines indicate the 95% CI of each study. The black vertical line represents the no-effect values, while the red dashed vertical line represents the overall effect size. The red diamond expresses the group-specific effect size, and the green diamond represents the global effect size. References: Ahn H (2020) [69]; Alshehri A (2020) [47]; Bamboat ZM (2014) [6]; Bu Z (2013) [55]; Chen J (2018) [66]; Chiu CT (2011) [52]; Chon HJ (2017) [65]; Dong X (2021) [72]; Efared B (2020) [22]; Gronnier C (2013) [76]; Guo CG (2015) [29]; Heger U (2014) [8]; Hsu JT (2016) [60]; Huh CW (2013) [27]; Hwang CS (2016) [33]; Imamura T (2016) [34]; Jiang CG (2011) [53]; Jiang H (2013) [56]; Jin X (2021) [73]; Jin EH (2015) [30]; Kang Sun H (2017) [38]; Kao YC (2019) [68]; Khan N (2020) [82]; Kim BS (2014) [28]; Kim YH (2016) [35]; Kim HM (2011) [25]; Kwak DS (2018) [41]; Kwon KJ (2014) [57]; Lee D (2018) [67]; Lee HH (2012) [54]; Lee IS (2017) [39]; Lee JH (2010) [23]; Lee SH (2015) [31]; Liu K (2017) [87]; Liu X (2015) [59]; Luu C (2017) [88]; Nakamura R (2019) [42]; Nam MJ (2010) [24]; Piessen G (2012) [79]; Postlewait LM (2015) [85]; Ryu DG (2019) [43]; Shim JH (2014) [58]; Taghavi S (2012) [84]; Tang CT (2020) [90]; Tang X (2016) [63]; Tong JH (2011) [26]; Voron T (2016) [81]; Wang Z (2015) [32]; Wang JB (2020) [71]; Wei Q (2020) [91]; Yoon HJ (2016) [36]; Zhang M (2010) [51]; Zhao B (2021) [78]; Zhao X (2021) [92]; Zhu ZL (2020) [44]; Zou Y (2020) [45]; Zu H (2014) [46].

**Figure 5 cancers-15-05191-f005:**
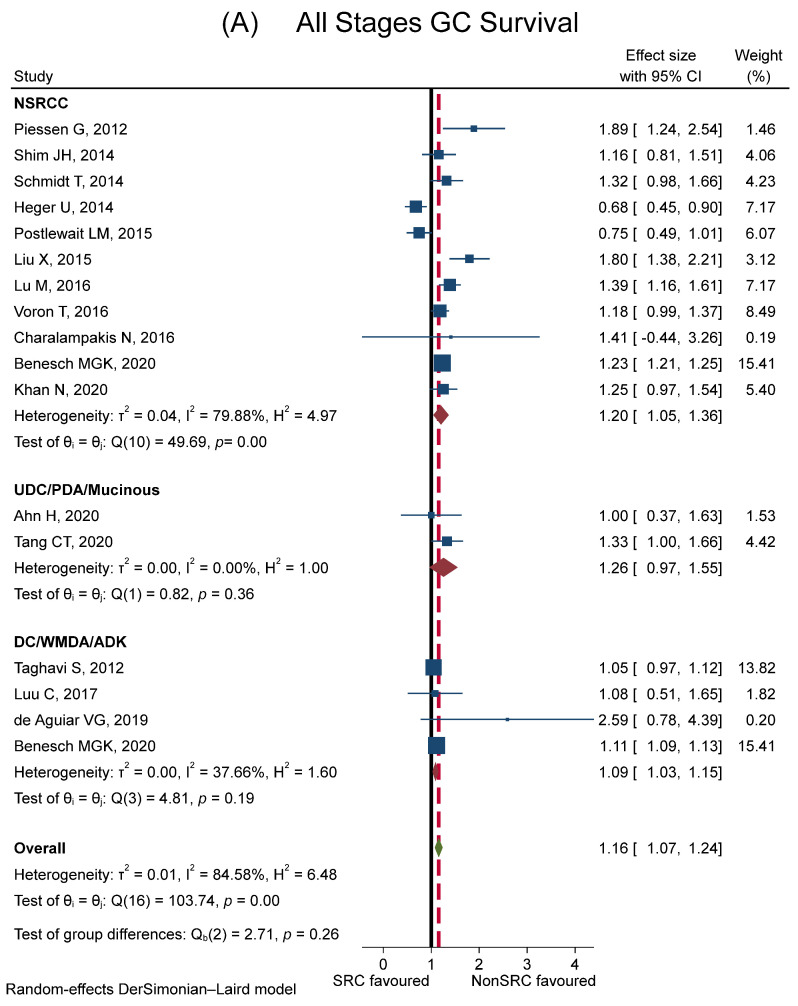
Forest plot showing meta-analysis results for survival (comparison of multivariable HR/RR). Plot (**A**): comparison between SRCC and other histologies (NSRCC, UDC/PDA/mucinous, DC/WMDA/ADK) in patients with all stages of gastric cancer. (**B**): comparison between SRCC and other histologies in patients with early, advanced, and metastatic gastric cancer. The blue square represents the size proportional to the weight of the study, and the horizontal lines indicate the 95% CI of each study. The black vertical line represents the no-effect values, while the red dashed vertical line represents the overall effect size. The red diamond expresses the group-specific effect size, and the green diamond represents the global effect size. References: Ahn H (2020) [69]; Alshehri A (2020) [47]; Benesch MGK (2020) [89]; Charalampakis N (2016) [86]; Cho JH (2015) [48]; Chon HJ (2017) [65]; de Aguiar VG (2019) [93]; Dong X (2021) [72]; Gronnier C (2013) [76]; Guo CG (2015) [29]; Heger U (2014) [8]; Hsu JT (2016) [60]; Imamura T (2016) [34]; Jiang CG (2011) [53]; Kao YC (2019) [68]; Khan N (2020) [82]; Lemoine N (2016) [77]; Liu X (2015) [59]; Lu M (2016) [62]; Luu C (2017) [88]; Men HT (2016) [49]; Piessen G (2012) [79]; Postlewait LM (2015) [85]; Riihimäki M (2016) [78]; Schmidt T (2014) [80]; Shim JH (2014) [58]; Shridhar R (2013) [83]; Taghavi S (2012) [84]; Tang CT (2020) [90]; Voron T (2016) [81]; Wang Z (2015) [32]; Zhao X (2021) [92]; Zu H (2014) [46].

**Table 1 cancers-15-05191-t001:** Information of the included studies sorted by continent and stage.

	Author	Year	Continent	Patients’ Source	Stage	n. SRCC	n. NSRCC	% SRCC
**1**	Efared B [22]	2020	Africa	Single institution	AGC	56	127	31%
**2**	Lee JH [23]	2010	Asia	Single institution	EGC	448	914	33%
**3**	Nam MJ [24]	2010	Asia	Single institution	EGC	720	1804	29%
**4**	Kim HM [25]	2011	Asia	Single institution	EGC	419	288	59%
**5**	Tong JH [26]	2011	Asia	Single institution	EGC	102	320	24%
**6**	Huh CW [27]	2013	Asia	Single institution	EGC	696	1512	32%
**7**	Kim BS [28]	2014	Asia	Single institution	EGC	345	1705	17%
**8**	Guo CG [29]	2015	Asia	Single institution	EGC	198	869	19%
**9**	Jin EH [30]	2015	Asia	Single institution	EGC	227	877	21%
**10**	Lee SH [31]	2015	Asia	Single institution	EGC	114	582	16%
**11**	Wang Z [32]	2015	Asia	Single institution	EGC	115	219	34%
**12**	Hwang CS [33]	2016	Asia	Single institution	EGC	233	317	42%
**13**	Imamura T [34]	2016	Asia	Single institution	EGC	190	556	25%
**14**	Kim YH [35]	2016	Asia	Single institution	EGC	927	368	72%
**15**	Yoon HJ [36]	2016	Asia	Single institution	EGC	930	2489	27%
**16**	Bang CS [37]	2017	Asia	Single institution	EGC	89	186	32%
**17**	Kang Sun H [38]	2017	Asia	Single institution	EGC	91	731	11%
**18**	Lee IS [39]	2017	Asia	Single institution	EGC	652	1185	35%
**19**	Horiuchi Y [40]	2018	Asia	Single institution	EGC	129	139	48%
**20**	Kwak DS [41]	2018	Asia	Single institution	EGC	331	206	62%
**21**	Nakamura R [42]	2019	Asia	Single institution	EGC	209	117	64%
**22**	Ryu DG [43]	2019	Asia	Single institution	EGC	233	143	62%
**23**	Zhu ZL [44]	2020	Asia	Single institution	EGC	287	230	56%
**24**	Zou Y [45]	2020	Asia	Single institution	EGC	146	177	45%
**25**	Zu H [46]	2014	Asia	Single institution	AGC	44	697	6%
**26**	Alshehri A [47]	2020	Asia	Single institution	AGC	219	1786	11%
**27**	Cho JH [48]	2015	Asia	Single institution	MGC	111	125	47%
**28**	Men HT [49]	2016	Asia	Single institution	MGC	40	17	70%
**29**	Choi JH [50]	2020	Asia	Single institution	MGC	171	516	25%
**30**	Zhang M [51]	2010	Asia	Single institution	All Stages	218	1221	15%
**31**	Chiu CT [52]	2011	Asia	Single institution	All Stages	505	1934	21%
**32**	Jiang CG [53]	2011	Asia	Single institution	All Stages	211	2104	9%
**33**	Lee HH [54]	2012	Asia	Single institution	All Stages	320	1056	23%
**34**	Bu Z [55]	2013	Asia	Single institution	All Stages	107	74	59%
**35**	Jiang H [56]	2013	Asia	Single institution	All Stages	288	80	78%
**36**	Kwon KJ [57]	2014	Asia	Single institution	All Stages	108	661	14%
**37**	Shim JH [58]	2014	Asia	Multicentric	All Stages	377	2266	14%
**38**	Liu X [59]	2015	Asia	Single institution	All Stages	138	1326	9%
**39**	Hsu JT [60]	2016	Asia	Single institution	All Stages	545	925	37%
**40**	Kong P [61]	2016	Asia	Single institution	All Stages	90	390	19%
**41**	Lu M [62]	2016	Asia	Single institution	All Stages	354	1845	16%
**42**	Tang X [63]	2016	Asia	Single institution	All Stages	260	244	52%
**43**	Wang Z [64]	2016	Asia	Single institution	All Stages	620	3310	16%
**44**	Chon HJ [65]	2017	Asia	Single institution	All Stages	1646	6021	21%
**45**	Chen J [66]	2018	Asia	Single institution	All Stages	62	179	26%
**46**	Lee D [67]	2018	Asia	Single institution	All Stages	176	588	23%
**47**	Kao YC [68]	2019	Asia	Single institution	All Stages	755	2216	25%
**48**	Ahn H [69]	2020	Asia	Single institution	All Stages	200	260	43%
**49**	Huang KH [70]	2020	Asia	Single institution	All Stages	181	260	41%
**50**	Wang JB [71]	2020	Asia	Single institution	All Stages	449	2893	13%
**51**	Dong X [72]	2021	Asia	Single institution	All Stages	254	3885	6%
**52**	Jin X [73]	2021	Asia	Single institution	All Stages	121	2045	6%
**53**	Zhao B [74]	2021	Asia	Single institution	All Stages	235	1656	12%
**54**	Bozkaya Y [75]	2016	Asia–Europe	Single institution	All Stages	142	51	74%
**55**	Gronnier C [76]	2013	Europe	Multicentric	EGC	104	317	25%
**56**	Lemoine N [77]	2016	Europe	Multicentric	MGC	57	146	28%
**57**	Riihimäki M [78]	2016	Europe	Swedish registry	MGC	82	736	10%
**58**	Piessen G [79]	2012	Europe	Single institution	All Stages	96	158	38%
**59**	Heger U [8]	2014	Europe	Single institution	All Stages	235	488	33%
**60**	Schmidt T [80]	2014	Europe	Multicentric	All Stages	221	516	30%
**61**	Voron T [81]	2016	Europe	FREGAT	All Stages	899	900	50%
**62**	Khan N [82]	2020	Europe	Multicentric	All stages	198	2302	8%
**63**	Shridhar R [83]	2013	North America	SEER	MGC	372	4200	8%
**64**	Taghavi S [84]	2012	North America	SEER	All Stages	2666	7580	26%
**65**	Bamboat ZM [6]	2014	North America	Single institution	All Stages	210	359	37%
**66**	Postlewait LM [85]	2015	North America	Multicentric	All Stages	312	456	41%
**67**	Charalampakis N [86]	2016	North America	Single institution	All Stages	62	45	58%
**68**	Liu K [87]	2017	North America	SEER	All Stages	4418	14877	23%
**69**	Luu C [88]	2017	North America	Single institution	All Stages	57	153	27%
**70**	Benesch MGK [89]	2020	North America	SEER	All Stages	17942	65218	22%
**71**	Tang CT [90]	2020	North America	SEER	All Stages	5265	752	88%
**72**	Wei Q [91]	2020	North America	SEER	All Stages	1751	7493	19%
**73**	Zhao X [92]	2021	North America	SEER	All Stages	3006	3673	45%
**74**	de Aguiar VG [93]	2019	South America	Single institution	All Stages	72	144	33%

Abbreviations: EGC: early gastric cancer; AGC: advanced gastric cancer; MGC: metastatic gastric cancer; FREGAT: French EsoGastric Tumours registry; SEER: Surveillance, Epidemiology, and End Results Program registry.

## Data Availability

The original contributions presented in this study are included in the article/online resources.

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
