# Peer review of "Clinical Features of Gastric Signet Ring Cell Cancer: Results from a Systematic Review and Meta-Analysis"

_cancers, 2023, doi:10.3390/cancers15215191_

Round 1

Reviewer 1 Report

Comments and Suggestions for Authors

I would like to commend the authors on a well written manuscript. This is a large meta-analysis of 74 studies investigating the prognostic significance of signet ring cell histology in gastric cancer on clinical outcomes including survival and nodal metastasis. They demonstrate lower risk of lymph node metastases in early stage disease but worse overall survival across all stages. The manuscript may be further considered with the edits and clarifications below: 

1. Were the authors able to screen duplicate retrospective reviews from the same institution? I.E. multiple Asian and SEER studies are cited - how did the inclusion criteria between these studies differ and does this have implications on the results of the analysis?

2. In the methods can the authors clarify the criteria used to define SRCC in this analysis? In the conclusion the most recent WHO criteria is cited (>90% SRCC type) - If this is reported in a subset of the studies does this new stratification change the prognostic effect?

3. In the discussion the authors comment on the higher rate of peritoneal carcinomatosis for SRCC, how does this rate compare between the other subtypes of gastric cancer?  

4. Was there criteria used to select studies that used modern staging techniques - EUS/PET/diagnostic laparoscopy? If not this should be commented as a limitation of the analysis. 

Minor Editing:

Line 280 - "without"

Line 302 - "sort of" avoid imprecise language

Line 331 - "mixing up" replace with "confusing" or "mistaking"

Line 349 - "smoots out" replace "is obviated", please clarify 

Reviewer 2 Report

Comments and Suggestions for Authors

The paper is well written, including a high number of patients and statistics. However, the subject is not particuarly interesting and results are a simple reporting of numbers without significance. According to this, the acceptance basically depends on editor’s decision.

Author Response

Please see the corrected version of the manuscript attached

Reviewer 3 Report

Comments and Suggestions for Authors

The manuscript looks very interesting however it can be improved by addressing the following issues:

1. In the Abstract, there are a lot of abbreviations without explanation of their full meaning that can be very unclear for the readers. 2. In the Introduction you mention the incidence of gastric cancer - but where in the world, in a specific country? Also, you cite a reference from 2019 for giving the incidence rate - which is quite old. Could you cite something more up-to-date? 3. Why for statistical analysis: Random-effects DerSimonian-Laird model was used? Were also other models checked? If so, which ones and how they performed? 4. What are the novel findings (different from already published articles) in this study? 5. Was the study registered in the database of systematic reviews like Prospero, etc.? Comments on the Quality of English Language

Minor English edits required.
